# Freshwater Management Discourses in the Northern Peruvian Andes: The Watershed-Scale Complexity for Integrating Mining, Rural, and Urban Stakeholders

**DOI:** 10.3390/ijerph20064682

**Published:** 2023-03-07

**Authors:** Daniel Mercado-Garcia, Thomas Block, Jheni Thalis Horna Cotrina, Nilton Deza Arroyo, Marie Anne Eurie Forio, Guido Wyseure, Peter Goethals

**Affiliations:** 1Aquatic Ecology Research Unit (AECO), Department of Animal Sciences and Aquatic Ecology, Ghent University, 9000 Ghent, Belgium; 2Centre for Sustainable Development, Department of Political Sciences, Ghent University, 9000 Ghent, Belgium; 3Facultad de Ciencias Sociales, Universidad Nacional de Cajamarca, Cajamarca 06003, Peru; 4Facultad de Ciencias de la Salud, Universidad Nacional de Cajamarca, Cajamarca 06003, Peru; 5Division of Soil and Water Management, Department of Earth and Environmental Sciences, KU Leuven, 3001 Leuven, Belgium

**Keywords:** gold mining, semi-structured interviews, mountain freshwater, discourse analysis

## Abstract

The Peruvian environmental action plan seeks headwaters protection as one of its integrated watershed management objectives. However, heterogeneous social and environmental conditions shape this freshwater management challenge at subnational scales. We have noticed different interpretations of this challenge. To map the debate, understand the diverse interpretations, and frame political choices, we conducted semi-structured interviews with institutional and non-institutional stakeholders for performing discourse analysis in an Andean watershed where mountaintop gold mining, midstream farmers, and the downstream Cajamarca city coexist. One discourse dominates the debate on protecting the freshwater supply and argues the importance of river impoundment, municipal storage capacity, and institutional leadership. The other two discourses revolve around protecting the mountain aquifer. The second discourse does so with a fatalistic view of headwaters protection and rural support. The third discourse partially shifts the debate towards the need for improving rural capacity building and (ground)water inventories. To understand evolutions in society, it is crucial to understand these three discourses, including the types of knowledge that actors present as legitimate, the attributed roles to all stakeholders, and the kinds of worldviews informing each discourse. The interaction among discourses could hinder integrated watershed management at worst or, at best, help inspire multi-stakeholder collaboration.

## 1. Introduction

Freshwater management for inland mining is expected to contribute to sustainable development [1]. However, the young environmental legislation and scientific advances in the mining sector, in combination with global phenomena such as climate change, social awareness, market fluctuations, and technological preferences, make mining sustainability a wicked problem [2]. The clarification of wicked social-ecological problems is helped by constructivist social assessments, which acknowledge that the human–environment system is defined, evaluated, and impacted by its actors within a given spatial and temporal context [3]. These actors, in turn, converge into coalitions defending their arguments, interests, worldviews, and practices around an environmental management problem. Thus, the social layer of information can be interpreted from several perspectives rather than from a single, often established perspective. This is important because dominant discourses are prone to overpowering and blocking resolutive alternatives or niche innovations [4].

In Peru, the Ministry for Environment has formulated a National Environmental Action Plan (PLANAA). The water sector’s goals include (i) integrated watershed management with a focus on headwaters protection, ecosystems, and sustainable management; (ii) guaranteeing the control of discharges in waterbodies; (iii) improving the water use and availability for the agricultural sector; and (iv) guaranteeing the full coverage of wastewater treatment and reuse in urban areas and increasing it in rural areas [5]. In addition to water, the PLANAA seeks to address problems of solid wastes, air quality, climate change, biological diversity, environmental governance, mining, and energy. Mining regions in Peru typically have a combination, or sometimes all, of the PLANAA challenges.

In the northern Peruvian Andes, the Yanacocha mine (MYSRL) exploits gold and silver ores at the headwaters of four watersheds, wherein two (Mashcon and Chonta) are sub-basins of the Amazon and the other two (Rejo and Honda) drain into the Pacific Slope [6]. Assessments downstream of the mine have found both positive (e.g., landowners increased income) [7] and negative impacts (e.g., human intake of heavy metals, land use planning lobbying, freshwater ecological impairment, and decreased phreatic level) [6,8,9,10]. Studies with no focus on mining, yet located in the MYSRL influence area, have reported anthropogenic pressures from logging, farming, and water abstraction for drinking and irrigation. These non-mining pressures resulted in either enhancement [11,12] or losses of ecosystem services [13,14,15]. While the main urban centre downstream of the mine (Cajamarca city) grew rapidly to accommodate the mining workforce, part of the population remained with unequal socio-economic opportunities [16]. Such a complex scenario shaped by nearly 30 years of large-scale mining operations constitutes a wicked challenge for the PLANAA objectives. Since stakeholders’ disagreement and knowledge scarcity/uncertainty characterise such wicked problems [17,18], we applied discourse analysis to help clarify the debate on freshwater management at the watershed level. The latter by addressing two research questions: How do the actors in the Mashcon watershed interpret the freshwater management challenges? Moreover, how influential can these interpretations be for freshwater management developments?

### Constructive Interpretation of Wicked Freshwater Management Challenges

A discourse can be defined as “a specific ensemble of ideas, concepts, and categorisations that are produced, reproduced, and transformed in a particular set of practices and through which meaning is given to physical and social realities” [19]. A discourse includes certain ideas but also excludes specific aspects from the debate, and it influences what is thought, seen, and performed within a social group. As such, discourse is thus a factor that can shape transitions [4].

Rather than opposing economic or engineering analyses, discourse analysis is an asset for addressing complex environmental problems [20] because it “does not look for truth—but rather at who claims to have truth” [21]. Ideally, an iterative deliberation among stakeholders defending the various discourses around a problem would enable consensus on overarching objectives. In practice, however, the persistence of top–down power relations, isolated scientific disciplines, inflexible practices, and social exclusion hinder the consideration of all stakeholders’ needs. Shedding light on this challenge, discourse analysis aims at mapping the discourses for understanding how the barriers to interaction emerge among groups of actors and, in return, how these barriers are confronted with potential outcomes [22].

Discourse analysis has been used to map and scrutinise freshwater management debates. For instance, the transition to a bio-based economy in the Netherlands encounters two interpretations for addressing water quality monitoring, either focusing on norm-compliant physicochemical quality or norm-flexible ecological functionality [23]. Likewise, the future view of wastewater treatment in the same country includes three discourses, focusing either on optimisation of centralised infrastructures, citizen awareness and decentralised systems, or emerging pollutants affecting public health [4]. The advantage of using a constructive–interpretative approach lies in its social–systemic informative nature, which is particularly important since citizens’ involvement and informal events are key to environmental decision-making [24].

## 2. Materials and Methods

### 2.1. Case Study

The Mashcon watershed (Figure 1), northern Peruvian Andes, hosts large-scale opencast mining in the headwaters, rural communities at midstream, and the Cajamarca city downstream. The city’s Mashcon River results from the confluence of the Grande River (GR) and Porcon River, wherein the latter originates from the unmined Mount Quilish. The GR is artificially recharged at the mountaintop by MYSRL to compensate for the headwaters’ loss due to mining operations. At midstream altitudes, freshwater is captured from the RG to make 70% of the Cajamarca city water supply. Conversely, rural inhabitants use the GR water for agricultural purposes only, whilst their domestic water comes from separate water springs.

A total of 28 watershed stakeholders were interviewed for this study, including mining, institutional, rural, and urban respondents. These consisted of six inhabitants of Cajamarca city and sixteen inhabitants of the rural homesteads of Llushcapampa Baja, Llushcapampa Alta, Llanomayo, Purhuay Bajo, Quishuar Corral, Tual, Aliso Colorado, and Cince las Vizcachas. The remaining six interviewees consisted of one representative each for the following institutions: Cajamarca’s water company (SEDACAJ), MYSRL, National Meteorology and Hydrology Service (SENAMHI), Cajamarca’s Regional Government (GORE), Municipality’s Management of Basic Sanitation, Natural Resources and Climate Change (municipality from here onwards) and the board of water users of the Mashcon (JUM). The interviews were conducted in 2018, except for a videoconference with the municipality in 2020 and MYSRL, which was interviewed in 2019.

### 2.2. Semi-Structured Interviews

Semi-structured interviews were conducted following ethical (i.e., requesting permission) and technical (i.e., avoiding influencing the response) considerations (see Appendix A). The interviews started with open questions about lifestyles or institutional roles. Freshwater-related information was obtained by alternating between spontaneous, essential, additional, and inquest questions (Table A1), including one last question about envisioning future scenarios. Each interview took between 15 and 47 min, with an average duration of 25 min. A corpus was built from the audio transcripts and inspected several times to identify excerpts for constructing the discourses. MS Word and MS Excel were used for reading and classifying the text.

### 2.3. Analytical Framework

For performing discourse analysis, we adapted the approach from Hajer [22] and Dryzek [20] to the local context. Past substantial changes in the Mashcon watershed were summarised in a historical background. The interviewees served as a source for constructing the meaning given to the freshwater’s physical and social realities. For analytical purposes, the discourses were represented in an ideal–typical way (or in its purest form), even though there were overlapping arguments in the debate. Each discourse’s definition of the freshwater management problems and solutions was contextualised in a storyline. A multi-level strategy was identified by distinguishing solutions at three levels, namely the roles of technology, institutions, and water users. The analysis also aimed at identifying who is seen as part of the problem and who is part of the solution.

The discourse elements (Table 1) were interpreted by considering the worldviews (“roots”) influencing the arguments and practices of the different social groups (“storylines”). These groups of actors were identified according to their convergence into a set of arguments for the concrete interpretation of the controversy (“discourse coalition”). Lastly, the discourse’s level of influence was interpreted in two folds, namely whether it dominates or not “the way a social unit conceptualises the world” (discourse structuration), as well as its influence on institutional arrangements and organisational practices (discourse institutionalisation) [4].

## 3. Freshwater Management in the Mashcon Watershed: Historical Background and Current Discourses

### 3.1. Historical Background of the Freshwater System

#### 3.1.1. The Establishment of Monocropping in Rural Areas

Before the gold mining boom in Cajamarca, the creation of a milk condensation plant in 1974 motivated lower-valley landowners to invest in irrigation and dairy cattle. Eventually, slope farmers shifted their economic activities as well, from sheep to cattle farming and from food crops to forage. With the 1990s expansion of cattle farming to neighbouring provinces, forage crops were being deemed advantageous for being drought and freeze resistant, in addition to guaranteeing a rural income thanks to the industrial milk demand. The latter was helped by the establishment of a second dairy factory in 1999 [26]. The FAO reported in 2006 that 75% of the Cajamarca region’s milk is produced in small farms of less than 14 cows each [27]. In the Mashcon watershed, one interviewee claims that rural families have no more than four or five cows each (Interview 9). Another one adds that before, they “used to grow everything, such as oca, potato, wheat and barley. But now the people are dedicated to cattle and grass. Grass is mostly to sell milk” (Interview 21).

#### 3.1.2. The Influence of Large-Scale Gold Mining

In 1992, the headwaters of the Mashcon watershed were given in concession to MYSRL. By 2002, the mining personnel grew to 7561 from the initial 517 workers [28], and with it, the population influx to Cajamarca city [29]. Oddly, the supreme decree granting the MYSRL concession (D.S. N° 014-1992-EM) had “no requirement that concessions are aligned with local government plans for development, land use, ecological zoning, or water management. Nor has there been consultation with local populations before the granting of concessions” [30]. The mining explosions and dewatering of open pits at the mountaintop have lowered the phreatic level. Consequently, MYSRL pumps and treats groundwater to ensure a required minimum discharge into the GR headwaters [6]. As additional measures of freshwater management, MYSRL built concrete irrigation canals, the San José reservoir, and smaller family-sized reservoirs, which have supplied irrigation since 2007 [31]. The mine addressed complaints of downstream sediment loads and flow reductions by building two dams for sedimentation, one in 2002 in the Rejo River and another one in 2004 in the GR. Moreover, a 2016 study found that MYSRL complies with the physicochemical freshwater quality norms in the dry season [25], thanks to an advanced mine water treatment system.

#### 3.1.3. Rural and Urban Water Sanitation and Human Health Implications

Rural water sanitation (RWS) evolved after evaluating two management models: the supply and the demand approaches, wherein the latter was validated. The supply model of the 1990s departed from the central government’s technological preference under cost-benefit reasoning. Once installed, water systems were left in charge of local actors, often yielding the infrastructures defective. Conversely, the 2000s demand approach worked upon request of the demand. The local government and communities participated in the technology decision-making process. The communities’ investment, capacity building, and workforce for rural water projects aimed to a gradual modification in habits and systems ownership. The local authorities and water users became responsible for keeping the RWS services in a sustainable state. Sustainability in this context was understood as the integration of infrastructure, operation and maintenance, primary health care, and sanitary education [32]. Since 2005, the General Law of Sanitation Services has stated that rural inhabitants shall manage, operate, and maintain the RWS services. This is through a Management Board of Sanitation Services (JASS) (D.S. N° 023-2005/VIVIENDA Article 4, numeral 18).

Contrary to the rural case, the urban water management capabilities were surpassed by the population dynamics. Since 1980, no additional water plant has been built to date (Interview 24). Between 1993 and 2007, Cajamarca city experienced a faster population growth (3.4 percent per year) than the national average of 2 percent per year [7]. Likewise, the urban wastewater treatment has failed dramatically, as explained on the SEDACAJ website: “The population growth and expansion of sewerage services in Cajamarca city have led to a surpass of the treatment capacity of the current waste stabilisation ponds, originating an organic overload and the proliferation of bad odours and consequent nuisances for the neighbourhood. This situation has induced SEDACAJ authorities to discharge part of the raw waste waters into the Mashcon River, contributing to the pollution of this water body” [33].

Water sanitation deficiencies in Cajamarca, such as having “running water only for a few hours a day”, are worse in impoverished areas of the city due to infrastructural deficiencies. Rural immigrants also struggle to access sanitary and health services [16]. A 2016 river monitoring campaign confirmed that the worst impacts from littering, riverbank invasion, and sewage discharges are in the downstream urban section of the watershed [25] (i.e., outside of the tourist centre). Moreover, there are concerns about water pollution caused by toxic mining chemicals in part of the population [7], as confirmed by past experiences of a few of the rural interviewees (see Section 3.2.2, Roots). Furthermore, Barenys et al. found in 2014 that the proximity of water sources to the gold mine directly correlates with the increased presence of arsenic in the drinking water [8].

### 3.2. Three Discourses on Freshwater Management

The precedent historical overview highlights the complexity of freshwater management, marked by mining impacts mitigation measures in upstream rural areas as well as social evolutions in both rural and urban areas. Our analysis of the semi-structured interviews identified three freshwater management discourses. While there were overlapping arguments in the debate, the ideal–typical representation follows particular roots that inform each discourse. The identified discourse elements are summarized in Table 2.

#### 3.2.1. Discourse 1: More Water Production and Education

The first discourse highlights the need for increasing the freshwater withdrawal and municipal storage capacity. River impoundment, water users’ education, and institutional leadership are prominent arguments in the debate. The storyline is “casted in concrete” in the form of reservoirs, dams, and irrigation canals built by MYSRL, as well as in the municipal water infrastructure of SEDACAJ. These two institutions, and the municipality, narrate the storyline.

##### Roots

The mine’s water treatment, storage, and downstream allocation constitute a compensation measure “validated by the Local Water Authority and the Ministry of Agriculture” (Interview 26). The National Water Authority (ANA) authorises MYSRL to withdraw groundwater at a rate of 601.8 L.s^−1^ and states that “the recharges maintain in equilibrium with the groundwater reserves, substituting the water volumes withdrawn from the exploitation pits sufficiently” [34]. The SEDACAJ representative acknowledges that freshwater management is a “comprehensive challenge” and that they are “very careful to not waste the liquid element” in the different process stages “because purifying the water costs a lot” (Interview 24). The MYSRL representative adds that water is managed carefully, “especially because of the constant supervision we have by the state institutions” (Interview 26). Another way to address the downstream communities’ concerns about mining impacts on the headwaters was the creation of a participatory Commission for the Monitoring of Canals (COMOCA) [6], whose groups of interests are “the boards of water users of the Chonta River and Mashcon River. Those rivers drain from the upper part” (Interview 26). Recently, different interpretations have emerged from the interaction between mining/institutional water managers and rural stakeholders. For instance, the communities’ reluctance to a large-scale water project is believed to have originated from their misinterpretation, as one says for the Chonta dam project: “people protested thinking it was a mining tailings pool. But it was not true, it is a reservoir” (interview 26). Moreover, the SEDACAJ and MYSRL representatives argue, respectively, that “in the highlands, there is a belief that water is originating from the higher parts. And I mean a belief because in reality, as it is known, the water cycle is a closed cycle” (Interview 26); “when it rains, it does not end up into the rivers, but it reaches down to the sea. Vaporization is the hydrological cycle” (Interview 24). Such argumentation is aligned with a narrative being put forward since 2007 for facing Cajamarca’s water crisis [9]. Namely, storing the rainy-season water to face the dry season scarcity and increasing the freshwater supply by river impoundment.

##### Storyline: Problems

The freshwater management challenges revolve around three core problems. First, water shortages in Cajamarca city are worsened by increasing droughts and population growth. A press release of SEDACAJ states that “the absence of rain is causing the service to be restricted to only hours” [35]. The city water demand was estimated at 250 L.s^−1^ in 2012, but only 60% is supplied by SEDACAJ in the dry season. From the 214 L.s^−1^ of freshwater withdrawn by SEDACAJ in the dry season, only 150 L.s^−1^ are billed to clients. Around 30% of the treated flow is lost by clandestine connections and leaks (e.g., 50% of the distribution infrastructure has a finished lifespan) [36]. The mining representative adds that rural parcelling increases the water demand since, for instance, “before a user had 10 hectares. Then he had 10 children and divided everything. That is why there are more users” (Interview 26). A second problem is the stagnation of water projects. The MYSRL representative sees a challenge when “establishing relationships with the communities and trying to get along well, which is not always possible. But at least we try”. That is why the Chonta dam project has been left to the GORE and “the municipality to do it” (Interview 26). The municipality interviewee explains that coordination has stopped because “the former manager sadly died” from COVID-19, likewise a technician and “another person in charge of the JASS”, and the interviewee adds that “the whole issue is being re-organised” since “no one has yet been hired” (Interview 27). A third major problem is the freshwater degradation by non-mining water users, particularly due to their deficient water culture (Interviews 24, 26, and 27) and economic conditions (Interview 27). The latter is in reference to the JASS because their fixed-monthly fee is “too little for the service … not enough to maintain an optimal water system” in some rural areas (Interview 27). Likewise, regarding urban users who have storage tanks, insufficient maintenance makes that “the water contamination takes place within their premises” (Interview 24).

##### Storyline: Future

The Cajamarca city water demand will mount to 390 L.s^−1^ in 2030 [36]. To face the increasing pressures on freshwater resources, one respondent imagines that “better technologies should be generated that allow us to use, reuse, water in a better way” (Interview 26). In parallel, “future generations need to be oriented on sustainable development, equity in the economic, environmental aspects, and technology” (Interview 27). Interviewee 24 believes that, next to inter-institutional efforts, “there must be a marked leadership there, which should be the GORE’s responsibility”. Additionally, regarding the need for increasing freshwater sources, the mining representative highlights the importance of increasing the municipal storage capacity by saying that “Arequipa and Cuzco are dry areas where it rains much less than here. However, they do not have the problems that we have in the dry season because they do the storage” (Interview 26).

##### Storyline: Strategy

A river impoundment technology has been proposed by MYSRL and the Association Los Andes de Cajamarca (ALAC) to solve the urban supply–demand mismatch. The Chonta River dam project (Interviews 24 and 26) would enable the production of 370 L.s^−1^ of tap water, in addition to benefiting agriculture with 74% of its dammed volume. Incremental amendments to the existing municipal infrastructure (e.g., increasing the capture pipe diameter or additional connections for distribution and reservoirs) are short-term measures [36]. The MYSRL representative adds that “Yanacocha stores water and constantly discharges it into the canals” … “those spring waters have the same characteristics as the water we discharge, class 3 water … Class 3 water is not bad, it complies with what the law says” (Interview 26).

Regarding institutional roles, the MYSRL representative explains that a “mining company like Yanacocha, likewise others in the north or south, is an important factor for there to be water, but it is not the actor. The actor is the state through its institutions” (Interview 26). An inter-institutional platform (GIR) organised by the municipality, and with ALAC in the secretariat, is addressing the protection of peri-urban riverbanks and the rehabilitation of the collapsed waste stabilisation ponds for bird watching (Interview 27). The GIR, one says, “is a program that has several projects with the aim of providing more water for the city” (Interview 26). Regarding institutional arrangements, the SEDACAJ representative argues that “the legislative decree 1280 has been published. It indicates that the city should be integrated with the rural area. This means that water companies should be in charge of the JASS administration” (Interview 24). The mining representative adds that “if the law would say that Yanacocha, or any other mining company, must discharge drinking water, we would have to comply” (Interview 26).

Regarding water user roles, the municipality representative believes that “the biggest challenge is environmental education. People know …, but it is not yet applied. So, as long as there is no such a culture of water care, it is not only the responsibility of the provider company, but the responsibility of the user” (Interview 27). The mining representative explains that “it is a matter of culture of the neighbours who pollute” because as the water “goes down it becomes contaminated. Not by mining activities, but by the action of the same neighbours who pollute it”. Moreover, the representative adds that MYSRL coordinates with “canal users, agricultural producers and grassroots organizations” for implementing programs of water care and hygiene (Interview 26).

In practice, MYSRL shall comply with environmental norms, as well as social and economic viabilities of their mining operations (Interview 26). SEDACAJ shall cover the demand of their urban clientele (Interview 24). Moreover, the municipality’s management of basic sanitation, natural resources, and climate change authorises the JASS organization and provides them with technical support (Interview 27). In addition, ALAC administers the MYSRL donations as well as specific agreements for improving the public water service, with the aim to “facilitate processes linked to project execution under the terms of reference and products approved by the responsible institutions: the provincial municipality of Cajamarca and SEDACAJ” [36].

#### 3.2.2. Discourse 2: Fatalistic Water Springs Protection

The second discourse argues the plausibility of a worst-case scenario, where freshwater becomes scarcer and conflicts intensify. Hence, stricter freshwater surveillance and increased awareness among all stakeholders are needed. Protecting the headwaters, rural support, and religious beliefs are cornerstones of this discourse. The storyline is narrated by non-institutional stakeholders who reproduce their experiences with mining, authorities, or neighbours added to the debate.

##### Roots

Apocalyptic interpretations have been reported in rural communities of Cajamarca, related to the central government’s Programa Juntos. The proposed economic transfers to pregnant women and mothers of minors were seen as a “diabolical making” or an intention of “dangerous appropriation” [37]. Regarding MYSRL’s expansion projects, a 2012 protest in Celendín (located northeast of Cajamarca province) resulted in five dead peasants and several wounded after clashes with the police [38]. An important root of the conflict was that downstream communities did not believe that MYSRL would operate without causing pollution nor agreed with the strategy of replacing four high-altitude lakes with reservoirs [39] (p. 442). A rural interviewee explains what makes them trust their domestic water is that “the Mount Quillish is where the headwaters are born, aquifers of pure water. Most of us use that water” (Interview 4). Another one adds that “living here in the countryside is beautiful because it is free” (Interview 14). In the second discourse, both human and non-human hierarchies inform the interviewee’s explanations for freshwater flow reductions, as one summarises: “The cause is nature itself, climate change. We also have a powerful company in our headwaters, where they are also exploring the side part. Perhaps, it is possible that those impacts have reduced the flow from 57 to 41 litres per second. Only God knows. I am evangelic and, in the writings, it is said that the day where we will fight for water will come, and we see that happening already” (Interview 5). During our field visits, rural interviewees recalled cases of human skin rush (Interviews 3, 9, and 28) and cattle poisoning (Interviews 3, 5, 6, 10, and 13) after using the freshwater discharged by the mine, although it is not the case at present (Interview 6, 18, and 21). Today, the tarnished reputation of MYSRL and of authorities, climate variability, religious messages, among other things, contribute to a fatalistic view on freshwater scarcity.

##### Storyline: Problems

The freshwater management challenges depart from three core problems at least. First, rural inhabitants feel excluded by authorities (Interviews 1, 4, and 6) or overpowered by MYSRL (Interviews 3, 6, and 21). An urban interviewee agrees by saying that “the authorities fail to support the farmer, even though farming is the basis for the nourishing of humanity” (Interview 17). A farmer explains that “when we the communities make protests for water, we receive little support to our claims” … “when there is money, they say everything is alright” (Interview 6). A deputy mayor adds that “often the authorities, in times of political campaign, they arrive here and offer this and that to us. And in the end, as if the peasants were whatever, the peasants are not considered. I mean, it has happened several times and it continues!” (Interview 1). Another one refers to participatory monitoring practices by saying that “Yanacocha does it with COMOCA, but it is only known by them” (Interview 5). A second problem is the impact of mining on the functioning of freshwater. For interviewee 6, mining explosions produce earthquakes that deviate the water springs’ flow to other locations. Moreover, the interviewees reported having lost the recreational uses, quantity, and quality of freshwater (Interviews 1, 2, 3, 5, 6, 8, 9, 10, 19, 21, and 28). One explains, “Before in this river, that now looks more like a brook, there used to be trouts for feeding my family. But nowadays, there are no trouts due to mining” (Interview 9). Another says that “before, our water was pure, of translucent colour. But nowadays the water comes turbid, which is noticeable when collecting water in a container and observing the settling down of sediments at the bottom, and a bad smell due to the water treatment by Yanacocha” (Interview 2). Another one who believes that replacing the headwaters artificially “will never have the same natural value”, explains that “the cuncul, which is a frog-like animal but with its little big head and its tail, before there were many. Now not. You see them minimally nearby to the water springs” (Interview 21). The third problem is that “the population pollutes as well” (Interview 16). One remembers playing in the Mashcon river as a child, when ”it was fun to go into the river, the contamination was not that much. But now there is plenty of garbage” (Interview 20). A deputy mayor says that in Cajamarca city, “people drink a soda and throw it into the river. That is why the photos of the river full of garbage come out. And then they say that the mine is polluting, but it still gives a care” (Interview 16). Moreover, some rural users “steal” the water according to interviewees 1, 14, and 28, as one summarises saying: “What we saw when we were kids was a lot of water coming, because it rained more. And now the flow has diminished, and we are not conscious about it. People have stolen water from the canal and caused pipe breaks. This water is stolen for their irrigation and animals, due to the scarcity in the dry season” (Interview 14).

##### Storyline: Future

A deputy mayor believes that “the future will be more complicated … and for this we think that if there is that support, we would have prevention” (Interview 16). Another one says that “mining activities will expand and kill all the water springs that remain. The ones who will suffer the most will be the children of the future” (Interview 21). Several respondents share the view that water scarcity is a biblical premonition and a fate, “not only for our population, but worldwide. It’s written in God’s words” (Interview 1). In addition to converging into religious arguments (Interviews 5, 8, 12,15, and 21), another one states that “the water quantity and quality future will be a mortal war. That will be the end of humanity because there is no solution” (Interview 17).

##### Storyline: Strategy

No fundamental technological changes are needed, but support to “install modern equipment for chlorination” (Interview 2) as well as “technification of irrigation for the crops” (Interview 8). Freshwater source separation prevents the risk of contamination in rural areas (Interviews 1, 2, 4, 5, 6, 8, 10,12, 13, 14, and 21). The latter thanks to having piped water, or agua entubada, captured from water springs. Such a source, one says, “is the only one left. Nowadays we drink spring water, which we think has not yet been contaminated” (Interview 2). In urban areas, one interviewee consumes water that “comes from wells like underground terraces” (Interview 19). Another urban interviewee, who believes that “several people have died of liver fluke due to the water” in the countryside, prefers bottled water over the agua entubada because “they say that water is treated, but no analyses have been done” (Interview 17). One interviewee explains that the freshwater provision for non-mining users depends on the continuity of MYSRL’s operations (Interview 5). 

Regarding institutional roles, interviewee 13 says that ”there should be surveillance, and protection of the water springs” (Interview 13). The other two explain that, in addition to performing the chlorination, the JASS imposes fines on those who use the piped water for agricultural purposes (Interviews 1 and 21). One believes that authorities should “care for our health and the children of future generations. They should do the analyses, flow measurements in our water systems to have assurance in our waters” (Interview 5). In addition to economic support to rural water projects (Interviews 2 and 17), interviewee 4 explains that the training given by the medical post has prevented health problems related to water consumption. Interviewee 5 adds that MYSRL “also helps us with our household water”.

Regarding the role of rural water users, they rely on a learn-by-listening approach. As one explains: “We participate in capacity building and we are told that the river water is not for domestic use, but only for animals” (Interview 5). Another one adds that part-time jobs at MYSRL enabled him to learn about waste sorting and sharing this with the community because “we learn by listening from the speeches and apply it in our households” (Interview 16). Moreover, peasants “have septic tanks”, and “each family knows what to do with their wastes. Some garbage is buried” (Interview 8). Moreover, the importance of traditional ecological knowledge manifests among the rural arguments for protecting the headwaters. Andean grasslands or pajonales alto-andinos, as well as the aliso tree (Interviews 10 and 13), are known to attract and store water naturally (Interviews 1, 2, 6, 10, 13, 14, 17, and 21) since the grandparents’ times (Interview 1). Lastly, neighbours recommend that each other use the water carefully (Interviews 3, 4, 12, and 15) and, according to one, to “pray to God, because God created waters and seas, the earth and the mountains” (Interview 12).

#### 3.2.3. Discourse 3: Improve (Ground)Water and Rural Studies

The third discourse emerges from interviewees concerned over the disordered urban growth and lack of local studies (Interviews 7, 22, and 25). The storyline is narrated by representatives of the GORE, JUM, SENAMHI, and a rural respondent who, in summary, argues the need for improving the capacity building, informed decisions, and (ground)water inventories.

##### Roots

The 2002 decentralisation process transferred the function of land use planning from the national to the regional governments. In Cajamarca, the GORE identified areas of water potential within the MYSRL concession, resulting in “a power struggle or a controversy, which was both a technical and a political process” [9]. In the city, the deficient enforcement of an urban development plan has generated “a strong pressure on the territory, mainly on water resources” (Interview 22). While the headwaters loss due to mining was offset by providing artificial flows downstream, the current source of rural livelihood has some disadvantages. The latter, for example, because “twice as much water needed for potatoes is for grass. And it is necessary to water weekly, otherwise it dries quickly … For selling one litre of milk, we do not feed ourselves, our children, all malnourished. Quite a disgrace” (Interview 7). Moreover, a 2012 study in the Mashcon watershed found that human dietary intakes of arsenic, cadmium, and lead were higher closer to the mine than downstream [8]. Another study found excellent physicochemical water quality in the RG upstream section but also ecological impairment in contrast to the natural headwaters [10]. Today, the GORE, JUM, SENAMHI, and a rural interviewee acknowledge that freshwater management is a much more complex challenge beyond mining impacts. Regarding climate variability, the GORE representative believes that “there is still a lot to analyse here and have accurate data” (Interview 25).

##### Storyline: Problems

Three problems raise awareness on knowledge generation for understanding the freshwater management challenges. First, “the accentuation of climatic variability, which are very rainy periods greater than normal and very dry ones well below normal. That variability is becoming more intense and frequent”, in turn causing problems on freshwater distribution and soils (Interview 22). Second, uninformed decisions lead to resource squandering. One says that groundwater abstraction is being presented by certain municipalities as promising, although the urban groundwater reserve is uncharacterised (Interview 25). Another example is given by interviewee 7, who refers to a recent landslide caused by a collapsing reservoir, and highlights that the Ministry of Construction, Housing and Sanitation is expending a lot in fuel, machinery, and workforce to fix this problem; “why? because a previous study was not done” (Interview 7). Third, a flawed model of rural support. For instance, after participatory water monitoring practices, the water samples are sent to Lima, and “the results are communicated after almost three months. Everything that is there is explained to the people, but it is hardly understood” (Interview 23). Regarding agricultural support, one says that “the government provides potato seeds. Professionals make the function only to inform the number of people, hectares, seeds, potatoes, but they do not come to revise”. Likewise, MYSRL gave cows, sheep or guinea pigs to the peasants “but never gave training … Yanacocha gave guinea pigs to around 400 people. What are they now? around 10 or 20 people who maybe have a lot of guinea pigs. They give cows for letting it die later” (Interview 7). Another one adds that “young people do not have the vocation to be farmers; they do not seize water properly” (Interview 23).

##### Storyline: Future

Covering the water demand will be more difficult with an increasing population (Interviews 22, 23, and 25) and unstoppable climate change (Interview 22). One says it is necessary to “change much of our habits” and “search for alternatives that enable us to live with this phenomenon that is coming” (Interview 22). The increasing pressure on freshwater resources means “the city will be hit hardest” (Interview 22) since “the water we consume here in the city of Cajamarca is regulated in the watersheds” (Interview 25). Another says that “effective order is needed at all levels … In Cajamarca, and throughout Peru, the challenge is to leave informality and move to formality … This happens mostly in the urban area”(Interview 25).

##### Storyline: Strategy

Regarding the role of technology, an interviewee believes that freshwater protection is related to “making visible the problems that exist. Have a good knowledge of how our water sources are” (Interview 25). The JUM presentative explains that MYSRL “has a project”, which is “an agreement with ALAC and students of the last cycles of civil engineering of the Private University of the North. We seek to update the inventories to be able to give a good service and see the problems of the canals. There are surveillance systems as of 2017, where different public and private institutions participate in controlling the amount of water” (Interview 23). The SENAMHI representative explains that they “work at the watershed level as well. These issues of monitoring, climate analysis, at the level of districts, and watersheds” (Interview 22). Regarding the presence of excess chlorine or lime in the water, interviewee 23 believes that “perhaps our water does not need that measure, but another treatment”.

Regarding institutional roles, the SENAMHI and GORE representatives explain that their roles in water resources management are indirect by providing technical documents to the water authorities for them to make informed decisions. Such information is important “for risk management issues in water management” (Interview 22), as well as the determination of water potential areas. The latter was in collaboration with a Regional Technical Commission of 36 representatives, including the GORE (Interview 25). Regarding institutional arrangements, the GORE representative identified a loophole in the groundwater abstraction legislation because “this drainage is being treated equally as the drainage of wastewater” (Interview 25). For one interviewee, it is also important that MYSRL “complies with the standards of quality and quantity of water. Plant native plants, so that there is water” (Interview 22). Another recommends that “there should be a dialogue between countryside and city, authorities and population. Especially, making it comprehensible from authorities to the peasant and the investments that they make” (Interview 7). 

Rural water users should avoid clientelism. As interviewee 7 says, “that is badly taught, and the people keep on asking, and make protests for demanding more aid”. The same person adds that it is important to “share among both professionals and farmers and grow as one people”, in addition to recommending MYSRL invest in forming rural leaders by sending the best of the young generation to agriculture-leading countries such as The Netherlands or Israel (Interview 7).

## 4. Discussion

The interviewees’ interests, arguments, and practices on freshwater management were explored and classified into three discourses with their respective roots and convergent groups of actors. Therefore, the previous sections provided the answer to our first research question: the Mashcon watershed stakeholders have three interpretations of how to address the freshwater management challenges. Rather than focusing on the contrasts among discourses (Table 2), the aim is to clarify the debate for framing political choices. Addressing our second research question, we discuss each discourse’s capability of becoming dominant first, to then argue broader implications.

The level of influence of discourse can be analysed in two steps. On the one hand, the more concrete conceptualisations provided by the actors, the more structuration a discourse has and, thus, more chances of influencing the debate among storylines. On the other hand, institutional influence is reached if the definitions/arguments of discourse are translated into institutional practices or arrangements [4]. The dominant discourse is eventually successful at interacting with infrastructures, social groups, and norms [22]. This analysis helps understand why freshwater management is shaped by the sociotechnical regime, in addition to the influence of external shocks [40].

The first discourse reiterates the need for increasing freshwater sources and educating water users with strong institutional leadership. Such a view influences the debate throughout the watershed since water harvesting, storage, and moderate usage, in addition to freshwater surveillance by institutions, are important to the other two discourses as well. The strategies of water users’ education and incremental water infrastructure developments are reproduced by ALAC, which has collaborated since 2013 with the municipality to organise the GIR (a technical group of water and sanitation involving more than 20 organisational representatives from Cajamarca province) (Interview 27). The technological solution for the Cajamarca city water shortage (i.e., impoundment of the Chonta river) is institutionalised in the central government’s interest in “large hydraulic projects for the benefit of Agriculture” [9]. The role of mining water management finds institutional support in the political economy of Peru, fostering mining investments [29], in addition to the water norms allowing MYSRL and SEDACAJ operations. Although a vertical model is put forward (e.g., centralisation of rural water management), the storyline struggles to define the phenomenological and intercultural causes for the stagnation of water projects and what solutions may apply under a centralised model. Despite the latter, the objectives are concrete and build upon the existing infrastructure and an interinstitutional platform. Thanks to having a structured and institutionalised storyline, the first discourse influences the debate on freshwater management to a higher degree than the other two discourses.

The second discourse, narrated by non-institutional stakeholders, is based on a widespread fatalistic view of the future of water scarcity and the overpowering of rural freshwater users. These actors reproduce their negative experiences with mining or authorities into the debate, stressing that water-related social conflicts will increase. The consistency of such an argumentation throughout the watershed reflects a certain level of structuration, although medium rather than high. The latter because fatalistic arguments are discouraged by the first discourse, as the MYSRL explains: “I see that a future like the war for water is mostly projected. And all of that is a pessimistic sense. I believe that the human being is an extremely adaptable being” (Interview 26). The suboptimal conditions of Peruvian peasants’ agriculture have recently been appointed at the governmental level by approving the Agricultural Emergency Plan. The latter is in the framework of the Declaration of Emergency of the Agricultural and Irrigation Sector (D.S. N° 003-2022-MIDAGRI). This has led, for example, to a supreme decree obliging the milk industry to use fresh milk as raw material instead of imported powder ingredients (D.S. N° 004-2022-MIDAGRI), as well as a call for the restructuration of the AgroPerú budget (D.S. N° 007-2022-MIDAGRI). Despite these measures, the second discourse has no institutionalisation in the scope of the interviewees’ arguments. Thus, by using definitions disregarded by institutional actors, in addition to struggling to define the changes to be made in the current model, the second discourse hardly influences the debate or the large-scale freshwater management projects.

The third discourse’s storyline is narrated by a wood entrepreneur, the JUM, GORE, and SENAMHI representatives. These are indirect actors who highlight the importance of environmental knowledge generation (soils, groundwater, surface water, and climate) for making the freshwater management challenges visible. Although the third discourse lacks a strategy for integrating environmental knowledge generation with freshwater management measures, it is informed by weaknesses in making MYSRL’s freshwater management norm compliant while overlooking the urban vulnerability. The PLANAA seeks the integration of the GORE with the National System for Environmental Information (SINIA), but no concrete strategy for protecting the groundwater reserves has been given by the Peruvian government. Recently, guidelines for making inventories of groundwater sources have been published [41]. Moreover, the coalition struggles to define the logistical, technological, and budgetary implications of advancing environmental measurements. For instance, the territorial ordering of Cajamarca, as the GORE representative says, “is a study at the macro level with a scale of hundred thousand. So, the Mashcon basin, which is a micro level, we do not see that detail” (Interview 25). Therefore, the third discourse has less structuration than the second discourse and considerably lower institutionalisation than the first discourse.

The three discourses are primarily not contradicting each other, as shown by the recognition that freshwater protection is a challenge beyond mining impacts. Nonetheless, it is necessary to explore the barriers to interaction among groups of actors (Table 3). The first discourse diminishes the importance of a co-management approach with rural actors. Rural actors adopted the JASS responsibilities as part of their freshwater management traditions (i.e., adding to what was learned from their ancestors). The 2017′s legislative decree 1280 came 12 years after the JASS legislation. Thus, certain interpretations, such as the one about the legislative decree 1280 (i.e., that SEDACAJ should manage the JASS), by institutional water managers act as a barrier for integrating the rural strengths into broader water management plans. Farmers are expected to adopt urban models of water management eventually. However, the rural self-governance, agro-centric system is based on the Andean worldview. The Andean worldview is prevalent among indigenous communities in the Andean region of South America. It includes a strong emphasis on community and collective action, as well as a belief in the interconnectedness of all things and the role of supra-material forces in daily life. The transmission of knowledge from one generation to the next is also important [42,43]. Graddy-Lovelace [44] pointed out a comparable issue in Cuzco, Peru, regarding potato agrobiodiversity loss driven by monocropping. She argues that the extrication of the Andean worldview from dominant agrobiodiversity conservation methods “perpetuates colonialist devaluations of efficacious traditional ecological knowledge and epistemologies”.

Another study in Cuzco identified a fatalistic view on future climate change impacts on livelihoods in socioeconomically vulnerable groups of stakeholders [45]. Brügger et al. [45] argue that the “differences between various subgroups could be used to improve measures to adapt to the consequences of climate change by correcting misconceptions of the population and decisionmakers”. Based on our findings, we argue that correcting misconceptions is also important in freshwater management, particularly regarding the headwaters’ protection. For example, water springs are translated to manantiales in Spanish, but the term ojos de agua (i.e., water eyes) was often mentioned during our field visits, evincing the consideration of the freshwater ecosystem as “alive” rather than as an engineering-controllable system. The treatment of the Andean worldview as mere folklore rather than as an important element of analysis [37] is, therefore, to be avoided when implementing broader freshwater management objectives such as those from the PLANAA. Moreover, the proliferation of fatalistic arguments is likely detrimental to stakeholder collaboration. Neglecting such cultural aspects would hamper the implementation of integrated watershed management objectives or, in the worst-case scenario, lead to deadly protests caused by social polarisation [46]. 

One step forward in environmental policy was the updating of the 2004 legislation for environmental liabilities of mining investments (Law 28271), namely amending a loophole that enabled certain mining companies to avoid responsibility by resigning their mining rights. From 7104 unassigned environmental mining liabilities, the Peruvian government has remediated 1180 [47]. Unfortunately, end-of-the-pipe solutions are not without risks of failure. Freshwater management’s meaning-making and expert support have been problematic following a failure of MYSRL’s management of the headwaters. SEDACAJ claimed on the second of November 2022, based on report number 150-2022-DCC-GC/ EPS SEDACAJ S.A., that the RG flow had suddenly dropped from 200 to 90 litres per second with an unusual, unprecedented colour and blamed MYSRL for this. However, in the afternoon hours of the same day, a new official, conjoint, communication by MYSRL and SEDACAJ claimed that the issue had been caused by “hydric stress”. In addition to reports of dead trouts in one of the communities, such a change of versions of official documents remained questionable since the SENAMHI data showed that precipitations were above average in the weeks prior to the catastrophe [48]. The consideration of the second discourse’s arguments is presumably impaired by the power concentration in dominant institutional water managers. A comparable issue regarding the territorial planning of Cajamarca has been scrutinised in detail by Jeronimo et al. [9].

A window of opportunity for critical reforms in mining’s freshwater management norms, and probably for making the arguments of the second and third discourses relevant to decision-makers, is found in two of the five strategic guidelines of the National Action Plan on Business and Human Rights (PNA) 2021–2025 [49]. These guidelines, published by the Ministry of Justice and Human Rights, aim to develop “a culture of respect for human rights in the business environment under the framework of international standards” as well as the “design and strengthening of mechanisms to ensure that those affected by human rights violations have access to judicial, administrative, legislative, or other means of redress”. Such a public policy document finds its roots in the support from the United Nations (UN) Working Group on Business and Human Rights, the UN Human Rights Office, the International Labour Organisation (ILO), the Organisation for Economic Co-operation and Development (OECD), and the European Union. The implementation of the PNA would require a decentralised approach to account for the social heterogeneity of the Peruvian Andes. Moreover, sustainable management of local freshwater management challenges requires the adaptation of political-economic institutions to multiple discourses on how humanity relates to and depends on nature [50] and at multiple levels (i.e., the roles of markets, governments, technology, and social evolutions) [4,51]. 

Overall, freshwater management developments would benefit from increasing the emphasis on analysing how coalitions are formed and informed. What is being put forward as legitimate regarding the dynamics of meaning and knowledge is needed within policy networks, despite the unpredictability of the policy implementation outcomes [52]. The social-ecological systems in the Andes are varied in type. The policy arena requires the inclusion of such a high diversity as means of providing multiple solutions that contribute to sustainable uses of natural resources [53]. Inflexible practices (e.g., more centralisation) pose a risk of delaying alignment with trends in the global water sector (i.e., more modular systems) [40]. Federal systems, such as rural self-governance, offer a higher resolution of problem identification and reporting. Diagnostic and corrective measures could also improve if interaction with experts is fostered. The more stakeholder groups are combined in freshwater management projects, and the more their interpretations are ethically discussed (Table 3), the higher the chances of building social trust in large-scale freshwater projects.

Moreover, freshwater management challenges arising from different interests, interpretations, and practices among all the concerned stakeholders also take place at much larger scales than that of our study area and in other parts of the world. This is the case of Central Asia, which is conditioned by its historical background (i.e., Kazakhstan, Kyrgyz Republic, Tajikistan, Turkmenistan, and Uzbekistan were formerly the Soviet Union) and different socio-economic conditions at present, which in turn increase the social, normative, institutional, and infrastructural complexities for managing freshwater systems [54,55]. Likewise, transboundary river basins such as the Amazon (which our case study belongs to) or the Mekong River basin [56] can pose additional challenges due to complex power relations and different systems of governance, meaning that different discourses could be influencing freshwater management decisions at national, subnational as well as supranational levels. Furthermore, the presented methods can shed light on concrete freshwater monitoring and assessment, for instance, when designing integrated socio-environmental models in the context of sustainable development approaches [57]. This study is a relevant showcase in the context of social–ecological approaches that are worldwide still often lacking the integration of social sciences to tackle major issues related to climate change and biodiversity [58].

## 5. Conclusions

We found three ways of interpreting freshwater management: the incremental, fatalistic, and informative discourses, which are based on, respectively, institutional power concentration, rural overpowering, and the potential power of environmental knowledge generation. In order to understand evolutions in society, it is crucial to understand these three discourses, including the kinds of knowledge that actors put forward as legitimate, what roles they attribute to all stakeholders, and by what kinds of worldviews their discourses are informed. The need for honest dialogue among rural and urban, institutional and non-institutional stakeholders, as well as between professional and non-professional water managers, represents a multi-level challenge. Our findings also suggest that despite the high diversity of Andean societies, certain patterns emerge that require careful attention from authorities. A phenomenological understanding of the barriers to stakeholders’ interaction is relevant to broad freshwater management objectives, given the urgent need to accommodate the profound social–ecological changes induced by large-scale mining concessions, global change, and remnants of intellectual colonisation.

## Figures and Tables

**Figure 1 ijerph-20-04682-f001:**
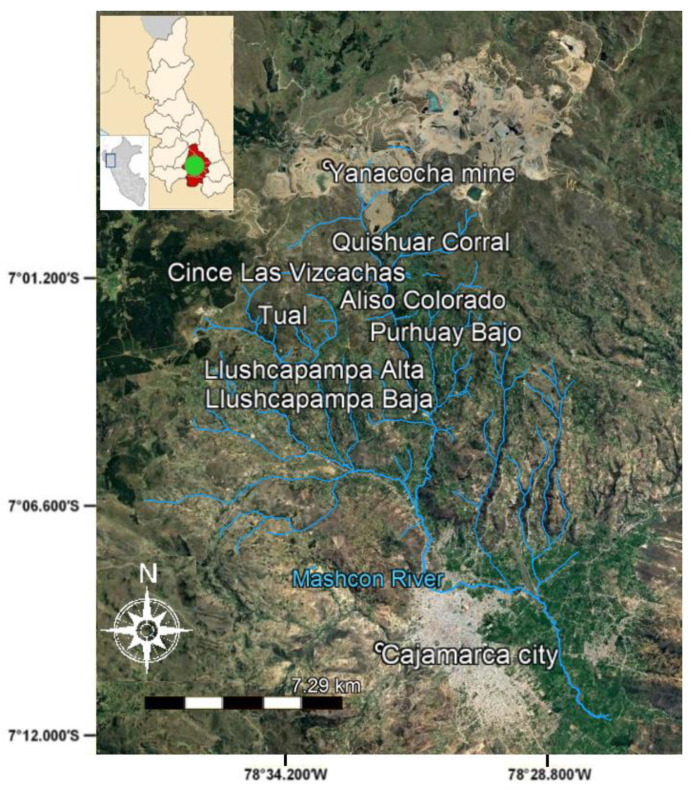
The Mashcon watershed and the locations relevant to this study. Adapted from [25].

**Table 1 ijerph-20-04682-t001:** Structure * of the analytical framework.

Discourse Elements	Definition
Roots	Worldviews embedded in a historical context
Storyline	What is being defined or put forward as legitimate by each group of actors
Storyline problem	As defined by a particular worldview
Storyline future	The projects of change and the imagining of a possible world
Storyline strategy	The suggested solutions, evincing the roles of technology, institutions, and water users
Discourse coalition	Groups of actors converging in their ideas, views, and actions
Influence	The discourse’s degree of success in shaping decisions and power concentrations

* Adapted from [4].

**Table 2 ijerph-20-04682-t002:** Summary of the three discourses.

	Discourse 1: More Water Production and Education	Discourse 2: Fatalistic Water Springs Protection	Discourse 3: Improve (Ground)Water and Rural Studies
Roots	Normative freshwater management “The water cycle is a closed cycle”	Distrust towards mining and authorities Human and non-human hierarchies	Technical and political struggle for zoning of water potential areas
Storyline problems	Drought and population growth Stagnation of water projects Flawed water culture and infrastructure maintenance in non-mining users	Rural exclusion and overpowering Freshwater degradation by mining “The population pollutes as well”	Climate variability accentuation (Water) resources’ squandering due to uninformed decisions Flawed model of rural support
Storyline future	City demand to be 390 L.s^−1^ in 2030 Better water technologies and storage Orientation of future generations Marked institutional leadership	Rural support for a complicated future Scarcer water, more mining, and intensification of water conflicts Potential water war	Find alternatives, change habits The city will be hit hardest Effective order needed at all levels
Storyline strategy: role of technology	River impoundment to produce water Incremental amendments of municipal water infrastructure and storage Normative mine water management	Chlorination and irrigation equipment Freshwater source separation Continuity of mine water management and downstream allocation	“Making visible the problems that exist” Update the water inventories Analysis at the watershed level Alternatives for water treatment
Storyline strategy: role of institutions	Main actors are the state institutions Inter-institutional platforms and mining-funded NGOs to facilitate water projects “SEDACAJ should manage the JASS”	Surveillance and protection of the remaining water springs Provide training and economic support	Indirect actors’ technical reporting Improve groundwater legislation Comprehensible dialogue between authorities and population
Storyline strategy: role of water users	Apply the environmental education Implement coordinated programs of water care and hygiene	Learn-by-listening, and applying it Traditional ecological knowledge Careful water usage and “pray to God”	Avoid clientelism “Grow as one people” Formation of young rural leaders
Coalition	Municipality, SEDACAJ, GIR, ALAC, and the mine	Non-institutional interviewees	GORE, SENAMHI, JUM, and one of the rural interviewees

SEDACAJ: Cajamarca water company; GORE: Regional Government of Cajamarca; GIR: interinstitutional platform; ALAC: Asociación los Andes de Cajamarca. JASS: rural management board of sanitation services; JUM: board of the Mashcon users; SENAMHI: national service of climate and hydrology.

**Table 3 ijerph-20-04682-t003:** Barriers to interaction and suggestions for stakeholders integration.

Groups of Actors	Barriers for Interaction	Suggestion for Integration
Discourse 1 (more water production) and discourse 2 (fatalistic protection)	Interpretation of legislation (e.g., legislative decree 1280) and rural actors sunk into resignation	Consider the Andean worldview as important element of analysis
Discourse 1 (more water production) and discourse 3 (improve water studies)	Logistical, technological, and budgetary implications	Changes in legislation about groundwater reserves
Discourse 2 (fatalistic protection) and discourse 3 (improve water studies)	No apparent barrier, as both groups of actors argue for the protection of the headwaters	Not applicable to the scope of this study
Among the three discourses	Different types of knowledge put forward as legitimate	Ethical deliberation of the different interpretations

## Data Availability

The raw data (audio recordings and corpus) are not publicly available due to privacy requirements but can be requested from the corresponding author if needed. Interview excerpts relevant to the study, and the anonymous list of interviewees, are contained in the Appendix A.

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
