# Peer review of "Freshwater Management Discourses in the Northern Peruvian Andes: The Watershed-Scale Complexity for Integrating Mining, Rural, and Urban Stakeholders"

_ijerph, 2023, doi:10.3390/ijerph20064682_

Round 1

Reviewer 1 Report

This is an interesting manuscript that considers the debate of the freshwater management in an Andean watershed where the mountaintop gold mining, midstream farmers, and the downstream Cajamarca city coexist. The article needs to make a clearer summary of the content of the interview and give more targeted opinions and suggestions. The paper should increase the

comparison and discussion with similar international studies, and summarize the solutions that can be used for reference, such as the problems of water resources utilization in Central Asia and Mekong River basin, as well as countermeasures and suggestions.

I think the authors found an interesting topic. I think the manuscript could be improved as follows:

1. Add the impact of human health in the background and section 3.1.3 which will be in line with the scope of this Journal.

2. Add the comparison of similar studies in the section of Discussion,such as the problems of water resources utilization in Central Asia and Mekong River basin.

Author Response

Comment 1. “The article needs to make a clearer summary of the content of the interview and give more targeted opinions and suggestions.”

Response 1.1: We agree with the Reviewer 1. Regarding the first part of the comment, a detailed guideline for the interviewing protocol was originally provided as Supplementary Material S2, but in Spanish. To fix this issue, we have created the new Appendix A to provide a summary of the guidelines for conducting the semi-structured interviews in English. Please see the new Table A1 in Appendix A. This new table has been cited within the main manuscript (section 2.2). It is important to mention that we protect the raw audios and transcripts of our interviews due to privacy protection reasons, although these can be specifically requested to the corresponding author if really needed. Therefore, our data availability statement has also been updated as follows:

“Data Availability Statement: The raw data (audio recordings and corpus) are not publicly available due to privacy requirements but can be requested from the correspondent author if needed. Interview excerpts relevant to the study, and the anonymous list of interviewees, are contained in the supplementary material S1 (Excel file).”

Response 1.2: Regarding the recommendation from the Reviewer 1 that more targeted opinions and suggestions are needed, we added a new table in the discussions section for communicating these aspects in a clearer manner. The new Table 3 is entitled “Barriers for interaction and suggestions for stakeholders integration”, and it contains a summary of key information from the discussion section that derives from the scope of the discourse analysis methodology. In other words, the targeted opinions and suggestions requested by the Reviewer 1 correspond, respectively, to the identification of the barriers for interaction and the suggestions for stakeholders’ integration. These aspects were originally mentioned in the discussion and are now summarized in a table. Please see the new Table 3 in the revised manuscript.

It is important to recall that although we collected “the roles of technology, institutions and water users” (section 2.3), these multi-level solutions correspond to arguments and practices from the interviewees, i.e., their own interpretations (Table 2). In this regard, Discourse analysis “does not look for truth, but rather at who claims to have truth” by using double hermeneutics (i.e., interpretation of the interpretations) . The latter, because “Discourse analysts are more likely to situate themselves politically, and to tell you contingently and conceptually where they think the power-relations are and how they are deployed.” (Carver, 2002). Furthermore, as stated in the first paragraph of our Discussion, “Rather than focusing on the contrasts among discourses (Table 2), the aim is to clarify the debate for framing political choices.” Therefore, the addition of Table 3 was made in line with the nature of the applied methodology, which is an interpretative-constructive approach that “aims at mapping the discourses for understanding how the barriers to interaction emerge among groups of actors and, in return, how these barriers are confronted with potential outcomes.” (section 1.1).

Comment 2. “The paper should increase the comparison and discussion with similar international studies, and summarize the solutions that can be used for reference, such as the problems of water resources utilization in Central Asia and Mekong River basin, as well as countermeasures and suggestions.”

Response 2:  We agree with the Reviewer 1. While a comparison of the discourses with international studies was not present in the original version of our manuscript, we agree that this is an important aspect to be addressed in the discussion section. For this end, we found five additional references that have been considered to improve the discussion section:

-Peña-Ramos, J.A.; Bagus, P.; Fursova, D. Water Conflicts in Central Asia: Some Recommendations on the Non-Conflictual Use of Water. Sustainability 2021, 13, 3479. (https://www.mdpi.com/2071-1050/13/6/3479 )

-Chatalova, L.; Djanibekov, N.; Gagalyuk, T.; Valentinov, V. The Paradox of Water Management Projects in Central Asia: An Institutionalist Perspective. Water 2017, 9, 300. (https://www.mdpi.com/2073-4441/9/4/300 )

-Sridhar, V.; Ali, S.A.; Sample, D.J. Systems Analysis of Coupled Natural and Human Processes in the Mekong River Basin. Hydrology 2021, 8, 140. (https://www.mdpi.com/2306-5338/8/3/140 )

- Forio, M.A.E.; Goethals, P.L.M. An Integrated Approach of Multi-Community Monitoring and Assessment of Aquatic Ecosystems to Support Sustainable Development. Sustainability 2020, 12, 5603. (https://www.mdpi.com/2071-1050/12/14/5603 )

- Maasri, A., Jähnig, S.C., Adamescu, M.C., Adrian, R., Baigun, C., Baird, D.J., et al. (2022) A global agenda for advancing freshwater biodiversity research. Ecology Letters, 25, 255– 263. Available from: https://doi.org/10.1111/ele.13931

The closure of our discussion has been updated by addressing the global importance of our study, prior mentioning of the issues present in Central Asia and the Mekong River basin because (and in line with what is important for doing discourse analysis) the freshwater management challenges in these regions are subjected to influences from their historical background (e.g. former Soviet Union, changes in governance systems), different interest among the concerned stakeholders (e.g., transboundary management of a watershed), as well as institutional challenges (e.g., inflexible or inefficient practices). Moreover, we consider important to mention that the presented methods can be a basis for concrete monitoring and assessment, as well as for the design of integrated socio-environmental models in the context of sustainable development approaches. Therefore, the updated discussion now ends with the following paragraph:

“Moreover, freshwater management challenges arising from different interests, interpretations, and practices among all the concerned stakeholders also take place at much larger scales than that of our study area and in other parts of the world. This is the case of Central Asia which is conditioned by its historical background (i.e., Kazakhstan, Kyrgyz Republic, Tajikistan, Turkmenistan, and Uzbekistan were formerly Soviet Union) and different socio-economic conditions at present, which in turn increase the social, normative, institutional and infrastructural complexities for managing freshwater systems [54,55]. Likewise, transboundary river basins such as the Amazon (which our case study belongs to) or the Mekong River basin [56] can pose additional challenges due to complex power relations and different systems of governance, meaning that different discourses could be influencing freshwater management decisions at national, subnational as well as supranational levels. Furthermore, the presented methods can shed light on concrete freshwater monitoring and assessment, for instance when designing integrated socio-environmental models in the context of sustainable development approaches [57]. This study is a relevant showcase in the context of social-ecological approaches that are worldwide still often lacking the integration of social sciences to tackle major issues related to climate change and biodiversity [58].”

Comment 3. “Add the impact of human health in the background and section 3.1.3 which will be in line with the scope of this Journal.”

Response 3:  We agree with the Reviewer 1. To mention health problems related to the study area and that are linked to water management, we updated the title of section 3.1.3 to “Rural and urban water sanitation and human health implications”. Therefore, section 3.1.3 now includes one additional paragraph that summarizes the water-related human health problems. For this end, it was not necessary to add new references since these aspects correspond to past experiences reported by the interviewees, as well as the findings by Barenys et al. (2014) (regarding heavy metals in the food chain), Bernet et al. (2002) (regarding concerns for water pollution by toxic mining chemicals), Steel (2013) (regarding the struggle to access sanitary and health services for lower income citizens of Cajamarca) and those from Mercado-Garcia et al. (2019) (regarding the severe urban pollution of rivers). The added paragraph is the following:

“Water sanitation deficiencies in Cajamarca, such as having “running water only for a few hours a day”, are worse in impoverished areas of the city due to infrastructural deficiencies. Rural immigrants also struggle to access the sanitary and health services [16]. A 2016 river monitoring campaign confirmed that the worst impacts from littering, riverbank invasion and sewage discharges are in the downstream urban section of the watershed [25], (i.e., outside of the touristic centre). Moreover, there are concerns of water pollution by toxic mining chemicals in part of the population [7], as confirmed by past experiences of few of the rural interviewees (see section 3.2.2.1). Furthermore, Barenys et al. found in 2014 that the proximity of water sources to the gold mine directly correlates with the increased presence of arsenic in the drinking water [8].”

Comment 4. “ Add the comparison of similar studies in the section of Discussion, such as the problems of water resources utilization in Central Asia and Mekong River basin.”

Response 4: We agree with the Reviewer 1. This recommendation has been considered and the modifications are explained in the above-mentioned Response 2.

Reviewer 2 Report

Review of the Article: “Freshwater management discourses in the northern Peruvian Andes: the watershed-scale complexity for integrating mining, rural and urban stakeholders” This is an interesting manuscript that deserves publication. Since the Peruvian environmental action plan seeks the headwaters protection as one of its integrated watershed management objectives. However, heterogeneous social and environmental conditions shape this freshwater management challenge at subnational scales. The authors have noticed different interpretations of this challenge. Then, the authors conducted a discourse analysis in an Andean watershed where mountaintop gold mining, midstream farmers, and the downstream Cajamarca city coexist.

The well-written manuscript only needs a few minor revisions to be more complete.

-          The abstract seems to be a very unclear methodology that was applied in this manuscript. Therefore, the abstract should be revised.

-          In the study area section, a map of locations such as the Mashcon watershed, the northern Peruvian Andes, mining areas, rural communities midstream, and Cajamarca city downstream could be included.

-          Why were 28 watershed stakeholders contacted for this study's interviews? Is there a single interviewee for every sub-watershed?

-          In addition, the authors should explain semi-structured interviews (why choose this method, advantages, and disadvantages compared to other methods).

-          More quantitative findings in research would be beneficial.

Author Response

Comment 1. “The abstract seems to be a very unclear methodology that was applied in this manuscript. Therefore, the abstract should be revised”.

Response 1: We agree with the Reviewer 2. To give more information about the applied methodology, the fourth sentence of the Abstract has been updated as follows:

“To map the debate, understand the diverse interpretations and frame political choices, we conducted semi-structured interviews to institutional and non-institutional stakeholders for doing discourse analysis in an Andean watershed where mountaintop gold mining, midstream farmers, and the downstream Cajamarca city coexist.”

Having to limit our abstract to 200 words, we hope that the above-mentioned modification of the abstract satisfies the Reviewer 2 expectations. Moreover, it is important to highlight that discourse analysis is a constructivist-interpretive approach that belongs to the phenomenological type of social science studies, and that it is of increasing interest and applicability to assess socially-complex environmental problems in a qualitatively manner. Also, discourse analysis does not focus so much on statistical generalizations, but rather on theoretical generalizations and transferability (See: Carminati L. Generalizability in Qualitative Research: A Tale of Two Traditions. Qualitative Health Research. 2018;28(13):2094-2101. doi:10.1177/1049732318788379). More details about the applied methodology are addressed in Responses 3 and 4 down below.

Comment 2. “In the study area section, a map of locations such as the Mashcon watershed, the northern Peruvian Andes, mining areas, rural communities midstream, and Cajamarca city downstream could be included.”

Response 2:  We agree with Reviewer 2. A new Figure 1 has been added to the manuscript in section 2.1 Study Area. This consists of a map of the Mashcon watershed and the locations relevant to the study. This figure has been adapted from our previous publication (as indicated in the figure caption), with the addition of labels indicating the names of the Yanacocha mine, the interviewed rural communities, and Cajamarca city. Please see the new Figure 1 in the revised manuscript.

Comment 3. “Why were 28 watershed stakeholders contacted for this study's interviews? Is there a single interviewee for every sub-watershed?”

Response 3: We interviewed at least one representative for each institution, and at least one stakeholder for each rural community. Rather than using a hydrographic differentiation for the subdivision of our interviewees, we aimed to have at least one interviewee per jurisdiction. Each of the rural communities have their own jurisdiction (i.e., each one is a homestead, with a self-governance organization and a deputy mayor). We try as much as possible to interview the deputy mayors or a president of the peasant patrol and, if needed, we interviewed more people until we reached a “saturation point” (see: Saunders, B., Sim, J., Kingstone, T. et al. Saturation in qualitative research: exploring its conceptualization and operationalization. Qual Quant 52, 1893–1907 (2018). https://doi.org/10.1007/s11135-017-0574-8 ) that enables the completeness of the discourse elements defined in Table 1. For example, in urban areas, after interviewing six people, we reached the saturation point of arguments that help constructing the discourses. In other words, interviewing more people would have been redundant since the arguments started to repeat over and over again among the different interviewees. The reasoning for selecting such a qualitative approach over a quantitative one is explained more in detail in Response 5.

Comment 4. “In addition, the authors should explain semi-structured interviews (why choose this method, advantages, and disadvantages compared to other methods).”

Response 4:   We agree with the Reviewer 2. We have added a new table in Appendix A, Table A1, to provide clearer explanations of how we conducted the semi-structure interviews. As stated by Keeffe et al. (2016) “semi-structured interviews are an effective and efficient method of collecting both qualitative and quantitative information for the assessment of drivers, behaviours, and their outcomes in a data-scarce region.” Considering that our methodology is qualitative (see also Response 5), the quantitative aspects of semi-structured interviews (or any other quantitative tool for social assessments) are not pertinent to the research design because discourse analysis “derives from a paradigm shift in philosophy”. Namely, it “represents an inversion of the scientific and commonsensical worldview inherited from the scientific revolution and empiricist philosophies of the last few hundred years”. The latter by moving “from taking a view about the world and its properties expressed through language, to a focus on language and its properties as such, and how the world is made for us from the meanings that language expresses” (Carver, 2002). In other words, semi-structured interviewing is the best and only suited method for the acquisition of qualitative information by using simple, open, questions in a comfortable social interaction without assuming the existence of a truth a priori, and without influencing the answers from the interviewees. The structural part of the “semi-structured” nature of the interviewing methodology consists in using questions addressing the matters important to the research objectives, as well as rephrasing essential questions to obtain more complete information. Likewise, inquest question were used for obtaining a personal opinion, sentiment, or critical thinking for a more profound exploration of the investigated matters. Please see the new Table A1 (new Appendix A) which summarizes the reasoning we used for conducting the semi-structured interviews.

Comment 5:  “More quantitative findings in research would be beneficial”

Response 5: To highlight the added value of the presented qualitative methods, we added the new Table 3 as means of communicating the key findings and recommendations in the scope of discourse analysis. While it is true that quantitative research helps to discover knowledge about a case study, it is also subject to aspects of sample size,  measurements bias, and instrumental capabilities. Rather than addressing our case study quantitatively, we used discourse analysis because it is a powerful qualitative method that allows researchers to gain insight into the complex ways in which language is used to construct meaning, convey power relations, and shape social interactions. As stated in section 1.1 of our manuscript “Rather than opposing economic or engineering analyses, discourse analysis is an asset for addressing complex environmental problems [20] because it ‘does not look for truth - but rather at who claims to have truth’ [21].” Being the methodology of the phenomenological type of social science studies, it is focused on meaning and interpretation (i.e, constructive-interpretive approach). Another strength of our methodology is that in enables flexibility in data collection and analysis, since it acknowledges that problems of high social complexity, such as freshwater management in the Mashcon watershed, would benefit from a deep understanding of the social and cultural contexts in which language is used. Moreover, discourse analysis acknowledges the subjective nature of interpretation, and the fact that different analysts may interpret the same data in different ways. This subjectivity is seen as an inherent strength of the method, as it allows for a more nuanced understanding of the complex social processes involved in language use. Furthermore, “Discourse analysts are more likely to situate themselves politically, and to tell you contingently and conceptually where they think the power-relations are, and how they are deployed.” (Carver, 2002).
